# Circulating Exosomal Integrin β3 Is Associated with Intracranial Failure and Survival in Lung Cancer Patients Receiving Cranial Irradiation for Brain Metastases: A Prospective Observational Study

**DOI:** 10.3390/cancers13030380

**Published:** 2021-01-20

**Authors:** Guann-Yiing Chen, Jason Chia-Hsien Cheng, Ya-Fang Chen, James Chih-Hsin Yang, Feng-Ming Hsu

**Affiliations:** 1Division of Radiation Oncology, Department of Oncology, National Taiwan University Hospital, Taipei 100, Taiwan; b96401107@ntu.edu.tw (G.-Y.C.); jasoncheng@ntu.edu.tw (J.C.-H.C.); 2Department of Medical Imaging, National Taiwan University Hospital Hsin-Chu Branch, Hsin-Chu 300, Taiwan; avonchen@ntuh.gov.tw; 3Graduate Institute of Oncology, National Taiwan University College of Medicine, Taipei 100, Taiwan; chihyang@ntu.edu.tw; 4Division of Medical Oncology, Department of Oncology, National Taiwan University Hospital, Taipei 100, Taiwan

**Keywords:** brain metastasis, whole brain radiotherapy, circulating exosomes, exosomal integrins, extracellular vesicles, biomarker

## Abstract

**Simple Summary:**

Brain metastases (BM) are the most common brain tumors in adults, and it remains a major complication in cancer patients. Exosomes or extracellular vesicles (EV) and integrins contribute to the development of BM, and exosomal integrins have been shown to determine organotropic metastasis. To our knowledge, this is the first clinical evidence demonstrating that high plasma EV integrin β3 level is associated with worse overall survival and intracranial control in BM patients undergoing whole brain radiotherapy, and supports the determination of organotropic metastases by exosomal integrins in the clinical setting. As utilization of whole brain radiotherapy for BM gradually declines in favor of local therapy such as neurosurgery and stereotactic radiosurgery (SRS), intracranial and distant brain control remain relevant because of the improved survival. Circulating EV integrin β3 may serve as a novel biomarker in predicting the progression of BM and survival.

**Abstract:**

Brain metastasis (BM) is a major problem in patients with cancer. Exosomes or extracellular vesicles (EV) and integrins contribute to the development of BM, and exosomal integrins have been shown to determine organotropic metastasis. We hypothesized that circulating EV integrins are able to influence the failure patterns and outcomes in patients treated for BM. We prospectively enrolled 75 lung cancer patients with BM who received whole brain radiotherapy (WBRT). We isolated and quantified their circulating EV integrins, and analyzed the association of EV integrins with clinical factors, survival, and intracranial/extracranial failure. Circulating EV integrin levels were independent of age, sex, histology, number of BM, or graded prognostic assessment score. Age, histology, and graded prognostic assessment score correlated with survival. Patients with higher levels of circulating EV integrin β3 had worse overall survival (hazard ratio: 1.15 per 1 ng/mL increase; *p* = 0.04) following WBRT. Multivariate regression analysis also showed a higher cumulative incidence of intracranial failure (subdistribution hazard ratio: 1.216 per 1 ng/mL increase; *p* = 0.037). In conclusion, circulating EV integrin β3 levels correlated with survival and intracranial control of patients with lung cancer after WBRT for BM. This supports that EV integrin β3 mediates a brain-tropic metastasis pattern, and may serve as a novel prognostic biomarker for BM.

## 1. Introduction

Brain metastases (BM) are the most common brain tumors in adults, and it is estimated that >20% of patients with cancer will develop BM [1]. As a result of improvements in diagnostic modalities, systemic control, and longer survival, the incidence of BM appears to be increasing [2]. Lung cancer is both the most commonly diagnosed type of cancer and the leading cause of cancer-related deaths [3]. Moreover, it is one of the types of cancer most frequently associated with BM [4,5]. Despite advances in multimodal treatment and systemic therapies, BM remains a major contributor to lung cancer-related mortality, while burdening patient quality of life and resources [6].

The prognosis of patients with BM depends on various factors, including age, performance status, primary disease control, presence of extracranial metastases, and treatment status [7,8]. In addition, non-small cell lung cancer (NSCLC) patients with activating mutations, such as epidermal growth factor receptor mutations (EGFR) or anaplastic lymphoma kinase (ALK) translocations, may have longer survival than wild-type patients [9]. The lung-molecular graded prognostic assessment (GPA) score is a revision of the disease-specific GPA to predict the survival of lung cancer patients with BM by incorporating molecular markers (EGFR and ALK alterations) along with the four established factors (age, Karnofsky performance status, extracranial metastases, and number of metastases) in patients with adenocarcinoma [10].

Cancer metastasis involves changes in cell–cell contact and cell–extracellular matrix adhesion [11]. Integrins, as the primary cell–matrix adhesion receptor, regulate tumor cell survival, stemness, and metastatic potential [12]. Colonization of a distant metastatic niche has been linked to the interaction between certain extracellular matrix proteins and integrins [13,14]. Integrin αvβ3, which is typically absent in normal brain stroma or vasculature, is abundant within angiogenic vessels in the peritumoral area of BM [15,16] and promotes intracranial metastatic growth [17]. On the other hand, integrin αvβ6-positive BM typically present with well-demarcated lesions and is associated with more favorable outcomes [18].

Exosomes or extracellular vesicles (EV) are small membrane vesicles (30–150 nm) containing functional biomolecules (i.e., proteins, lipids, RNA, and DNA) that can be transferred horizontally to recipient cells [19]. It has been shown that the secretion of EV by cancer cells can lead to pre-metastatic niche formation and subsequent metastasis [20,21,22], and exosomal integrins have been linked to specific organotropic metastatic patterns [23]. As surgical excision or biopsy of multiple BM is often impractical in clinical settings, the evaluation of circulating EV and associated integrins as biomarkers is attractive and may serve as a potential surrogate for an overview of the tumor integrin “landscape” that may influence the outcome. We hypothesized that circulating EV integrins are able to influence the failure pattern (intracranial versus extracranial) in patients treated for BM, and may serve as a novel biomarker independent of disease-specific GPA and other clinical factors for the prediction of survival. In particular, we proposed that circulating EV integrin β3 could be a potential prognostic biomarker regarding the development and progression of BM, as well as survival.

## 2. Results

### 2.1. Patient Characteristics and Outcomes

Between March 2015 and July 2019, 75 patients with BM originating from primary lung cancer were enrolled and received whole brain radiotherapy. The patient characteristics are summarized in Table 1. The median age was 61 years, and the majority of patients (76%) had ≥5 BM. The most common histology was adenocarcinoma (81%), and four patients had small-cell carcinoma primaries, typical for the East Asian lung cancer cohort. In terms of GPA score, 16%, 44%, 36%, and 4% had a score of 0–1.0, 1.5–2.0, 2.5–3.0, and 3.5–4.0, respectively. Six patients had received prior stereotactic radiosurgery to brain metastases. The median time from diagnosis of BM to whole brain radiotherapy (WBRT) was 102 days (range: 11–1636 days). Forty-nine patients (65%) were treated with conformal volumetric-modulated radiation therapy. At the median follow-up at 8.0 months (range: 0.7–52.1 months), death or intracranial progression had occurred in 76% of patients.

The median OS, IC-PFS, and EC-PFS for the entire cohort were 14 months (95% confidence interval [CI]: 6.6–21.6 months), 7.9 months (95% CI: 7.1–8.7 months), and 5.1 months (95% CI: 4.1–6.1 months), respectively (Figure 1). Patients with adenocarcinoma primaries had significant longer OS than those with non-adenocarcinoma primaries (median OS: 17.1 months vs. 4.2 months, respectively; *p* = 0.003) (Figure 2A). In addition, patients with higher GPA scores also had longer survival times (median OS: GPA score 0–1.0: 4.2 months; 1.5–2.0: 9.5 months; 2.5–3.0: 17.1 months; 3.5–4.0: 26.4 months; *p* = 0.005) (Figure 2B). There was a trend toward the worse survival with advancing age (hazard ratio [HR]: 1.32 per 10-year increase; *p* = 0.123).

Similarly, patients with adenocarcinoma primaries showed significantly longer IC-PFS than those with non-adenocarcinoma primaries (median IC-PFS: 8.0 months vs. 3.5 months, respectively; *p* = 0.002). Higher GPA scores were also associated with better IC-PFS (median IC-PFS: GPA score 0–1.0: 3.6 months; 1.5–2.0: 8.0 months; 2.5–3.0: 9.1 months; 3.5–4.0: 19.8 months; *p* = 0.002), and there was a trend toward worse IC-PFS with advancing age (HR: 1.32 per 10-year increase; *p* = 0.104). In contrast, EC-PFS was longer in patients with adenocarcinoma (median EC-PFS: 5.6 months vs. 2.4 months; *p* = 0.02) and younger patients (HR: 1.38 per 10-year increase; *p* = 0.026), but only trended toward better EC-PFS with higher GPA scores. The survival outcomes did not differ in relation to sex, number of BM, or presence of leptomeningeal carcinomatosis (Appendix A).

### 2.2. Expression of Circulating Extracellular Vesicles Integrins

Our literature review identified two subtypes of circulating EV integrins (integrin β3 and integrin β6) of particular interest. The median expression level of integrin β3 was 1.668 ng/mL (range: 0–38.501 ng/mL; 1st quartile: 0.955 ng/mL; 3rd quartile: 2.692 ng/mL), and the median expression levels of integrin β6 were 2.024 ng/mL (range: 0.654–13.814 ng/mL; 1st quartile: 1.373 ng/mL; 3rd quartile: 3.036 ng/mL). The medians of circulating EV integrin β3 and integrin β6 concentrations did not differ in relation to histology, number of BM, GPA score (Figure 3), age, or sex (not shown). Extreme outliers (integrin β3: 38.501 ng/mL and integrin β6: 13.814 ng/mL), both markedly higher than the 3rd quartile plus five-fold their respective interquartile ranges, were identified. Interestingly, both outliers were noted in the same patient (a 50-year-old male who had adenocarcinoma primary, >10 BM, and GPA score of 2.5). This particular patient experienced extracranial progression and subsequent death 8 months after receiving WBRT, without evidence of intracranial failure.

### 2.3. Expression of Circulating EV Integrins Independently Predicted Outcomes for BM

High levels of circulating EV integrin β3 correlated significantly with worse OS (HR: 1.15 per 1 ng/mL increase; 95% CI: 1.01–1.32; *p* = 0.04). The maximally selected log-rank statistics suggested that the optimal cut-point was 1.818 ng/mL, and the resultant high integrin β3 level group (>1.818 ng/mL) demonstrated shorter OS compared with the lower integrin β3 level group (≤1.818 ng/mL) (median OS: 6.5 vs. 18.9 months, respectively; HR: 2.12; 95% CI: 1.16–3.87; *p* = 0.01, Figure 4A). On the other hand, circulating EV integrin β6 did not correlate with OS (HR: 1.06 per 1 ng/mL increase; 95% CI: 0.869–1.29; *p* = 0.56). The univariate analysis of prognostic factors associated with OS were shown in Appendix A. On multivariate Cox regression analysis, age, histology, GPA score, and circulating EV integrin β3 levels were significantly associated with OS, whereas sex and integrin β6 levels did not show significant correlations (Table 2).

The 6-month and 1-year cumulative incidence of intracranial failure in the high integrin β3 level group were 39.0% and 61.7%, respectively, compared to 4.8% and 36.5% in the low integrin β3 level group (*p* = 0.036, Figure 4B), with death as the competing risk. The cumulative incidence of intracranial failure did not differ in relation to integrin β6 levels (*p* = 0.68). The univariate competing risk analysis of prognostic factors associated with intracranial failure were shown in Appendix A. On the multivariate competing risk regression model, only integrin β3 correlated with intracranial failure (subdistribution HR: 1.216 per 1 ng/mL increase; 95% CI: 1.012–1.46; *p* = 0.037), while the GPA score demonstrated correlation with the competing cause (death without intracranial failure), with age and histology exhibiting a non-significant trend (Table 3). Sex and integrin β6 levels did not show a significant correlation. In contrast, the multivariate competing risk regression model of extracranial failure did not reveal significant correlation with any of the aforementioned factors (Table 3).

## 3. Discussion

We demonstrated that circulating EV integrin β3 is associated with the clinical outcomes of BM in patients with lung cancer. Patients with high plasma EV integrin β3 expression had worse OS and intracranial control after WBRT. To our knowledge, this is the first clinical evidence indicating that EV integrin β3 is a novel biomarker for predicting the progression of BM and poor survival, and supports our hypothesis that the determination of organotropic metastases using EV integrins is possible.

Our results suggest that the circulating EV integrin β3 or β6 levels are independent from age, sex, histology, number of BM, or GPA score. The relationship between the tumor cellular expression of integrin (either in primary tumors or metastases) and EV integrin levels has not been established yet; however, evidence supporting integrin transfer via tumor cell-derived EV is growing [24,25,26]. Surgical pathological data have demonstrated that the majority of NSCLC express the αv-integrins αvβ5 and αvβ6, and to a lesser extent αvβ3 and αvβ8 [16]. However, in specimens from primary tumors, these integrins were not correlated with tumor proliferation, nodal spread, or survival [27]. Overexpression of αv-integrin had been shown to enhance the cell migration rate and lead to increased BM in a murine model [28]. As our study enrolled patients with pre-existing BM, the distribution of EV integrins was expected to be different from that observed in patients without BM at baseline.

The positive correlation between increasing EV integrin β3 levels and early intracranial failure after WBRT attested to the tendency of BM to occur in patients with high EV integrin β3 levels. Accumulating evidence on exosome-mediated organ-specific conditioning has shed light on Stephen Paget’s “seed-and-soil” hypothesis [29], which was proposed 131 years ago. In the landmark article published in *Nature*, Hoshino et al. demonstrated that the tumor exosome integrins α6β4/α6β1 and integrin αvβ5 were associated with lung and liver metastasis, respectively, while integrin β3 was present in exosomes isolated from brain-tropic cells [23]. Another recent study also showed that prostate cancer cells transferred integrin αvβ3 via EV, and the recipient cells exhibited a pro-metastatic phenotype [25]. Our results corroborated these findings in the clinical setting. In addition, a study also showed that tumor exosomal cell migration-inducing and hyaluronan-binding protein (CEMIP) can promote BM [30], highlighting the complexity of cross-talk between different molecules.

Furthermore, our study demonstrates that increasing circulating EV integrin β3 levels are associated with worse survival. The median survival of 14 months was in line with that noted in earlier cohorts, considering the distribution of GPA and the extent of BM requiring WBRT [10]. However, both the survival curves and cumulative incidence of intracranial progression between the high and low integrin β3 level groups converged as follow-up time exceeded 18–24 months. This observation indicated the generally poor prognoses of those patients and potential phenotypic changes in metastatic behavior later in the disease course. The EV microRNA (miRNA) profile, specifically the tumor suppressor miRNA let-7f, has also been shown to correlate with survival in patients with lung cancer [31]. Several EV miRNAs have also been linked to an increased incidence of metastases [32,33], suggesting multiple pathways through which EV could impact survival.

In contrast, EV integrin β6 levels were not associated with either a specific failure pattern or survival in our study. The upregulation of integrin ανβ6 expression is associated with poor prognosis in many types of cancer [34]. However, data concerning integrin β6 have been less clear, as surgical pathological data from primary NSCLC did not show any evidence of worse outcomes with integrin ανβ6 expression [27]. Notably, survival appeared to be better in non-squamous NSCLC patients with BM expressing integrin ανβ6 [18]. EV integrin β6 levels may be an indicator of worse tumor biology [34], but we speculate that other factors, such as EV integrin β3, may dominate the development and progression of BM.

As utilization of WBRT for limited BM gradually declines in favor of local therapies such as neurosurgery and stereotactic radiosurgery (SRS), intracranial and distant brain control remain relevant because of the improved survival times, and have become a concern without the use of WBRT [35,36]. A secondary analysis of the Japanese Radiation Oncology Study Group 99-1 randomized trial suggested that adjuvant WBRT after SRS may improve the survival rates in NSCLC patients with favorable prognoses [37], emphasizing the importance of intracranial control in certain patients with expected long-term survival. Metrics such as brain metastasis velocity after initial SRS had also been introduced [38]. EV integrin β3 may as well serve as a novel biomarker in predicting the development of BM and survival. Analyses of an independent SRS cohort and additional EV component repertoires are mandatory and currently underway, and our study provides evidence that circulating EV integrin signatures may aid clinical decision-making for patients with BM. Those with high circulating EV integrin β3 plus additional poor prognostic factor (e.g., low GPA score) may consider short-course WBRT or hospice care. An alternative systemic therapy with superior central nervous system (CNS) penetration to overcome blood-brain barrier for intracranial control as well as a more frequent CNS imaging follow-up schedule for early salvage therapy might be required for patients with increased circulating EV integrin β3.

The potential limitations of our study include the following: (1) The absence of pathological specimens to determine the relationship between the EV integrin levels and cellular expression of integrins makes its source a matter of debate; (2) insufficient data to analyze changes in EV integrin levels upon recurrence; and (3) possible inconsistencies in EV isolation and integrin quantification, which may be inherent as standardized protocols and reference levels were lacking at the time of assessment, thus limiting generalizability.

## 4. Materials and Methods

All patients were pathologically and/or cytologically diagnosed with radiographic evidence of BM, and provided informed consent at the time of enrollment. The present study cohort were treated with WBRT at the discretion of the attending radiation oncologist. In accordance with contemporary guidelines, radiotherapy with 30 Gy in 10 fractions was delivered with a linear accelerator using 6 MV photons. Two-/three-dimensional or conformal volumetric-modulated radiation therapy techniques with or without hippocampal avoidance were applied. Patient and tumor characteristics, including disease-specific GPA [8]/lung-molecular GPA [10], age, Karnofsky performance status, extracranial metastasis, number of BM, and driver mutation status were recorded at baseline. Blood samples were collected in the tubes containing anticoagulant EDTA prior to radiotherapy according to the protocol. Patients were followed up at 1, 2, 4, 6, 9, and 12 months, and every 3 months thereafter, until unequivocal intracranial progression, death, or patient withdrawal (whichever came first). Intracranial response was assessed according to the Response Assessment in Neuro-Oncology Brain Metastases criteria [39].

The EV were isolated from 1mL of stored plasma by size exclusion purification using Exo-Spin^TM^ Midi Columns (Cell Guidance Systems, St. Louis, MO, USA) according to the protocol provided by the manufacturer. The circulating EV integrins β3 and β6 were subsequently measured with an enzyme-linked immunosorbent assay (ELISA) using a commercially available human protein sandwich enzyme immunoassay kit (category numbers AE37655HU & AE37642HU; Wuhan Abebio Science Co., Ltd., Wuhan, China). Concentrations of integrins in EV were determined against a standard curve.

Overall survival (OS) was measured from the day of enrollment to death. Intracranial progression-free survival (IC-PFS) and extracranial progression-free survival (EC-PFS) were calculated from enrollment to death or until radiographic evidence of progression and/or recurrence of BM or leptomeningeal carcinomatosis, and extracranial progression, respectively. Patients without a known date of progression were censored at the time of the last follow-up. Correlations between the medians of circulating EV integrin concentration of subgroups were assessed with the Kruskal–Wallis test. The correlations between survival and EV integrin concentrations were assessed with univariate Cox proportional hazard regression. Extreme outliers were identified using Grubb’s method [40], and remained within their respective dichotomized categories for robustness of the analyses. However, they were trimmed in respective continuous variables to avoid bias in the regression analyses. As clinically relevant thresholds have not been established, we utilized maximally selected log-rank statistics to define cut-points according to differences in survival [41]. For comparison, we dichotomized graded prognostic assessment scores into high (2.0–4.0) and low (0–1.5). The groups were compared using Pearson chi-squared and Fisher’s exact tests, as appropriate. Kaplan–Meier analysis was used to estimate OS, IC-PFS, and EC-PFS, and these finding were compared between groups using log-rank tests. The cumulative incidence of intracranial and extracranial failure was assessed using competing-risks analyses with death as the competing risk, and groups were compared using Gray’s method [42]. Multivariate regression analyses were performed using the Cox proportional hazards model for OS, and subdistribution hazards model proposed by Fine–Gray [43] for intracranial and extracranial failure, considering death as the competing risk. Statistical analysis was performed with the IBM SPSS, version 25.0 (IBM Corp., Armonk, NY, USA) and R, version 3.6.0 (Free Software Foundation, Boston, MA, USA) with packages survival, maxstat, and cmprsk.

## 5. Conclusions

This prospective study identified circulating EV integrin β3 as a potential novel BM-specific biomarker to predict the outcomes after WBRT in patients with lung cancer. Patients with high circulating EV integrin β3 levels have worse intracranial control and poor survival outcomes, independent of the graded prognostic assessment for lung cancer using molecular markers. The translational evidence suggests that EV abundant in integrin β3 are brain-tropic in lung cancer. This finding requires validation in an independent cohort.

## Figures and Tables

**Figure 1 cancers-13-00380-f001:**
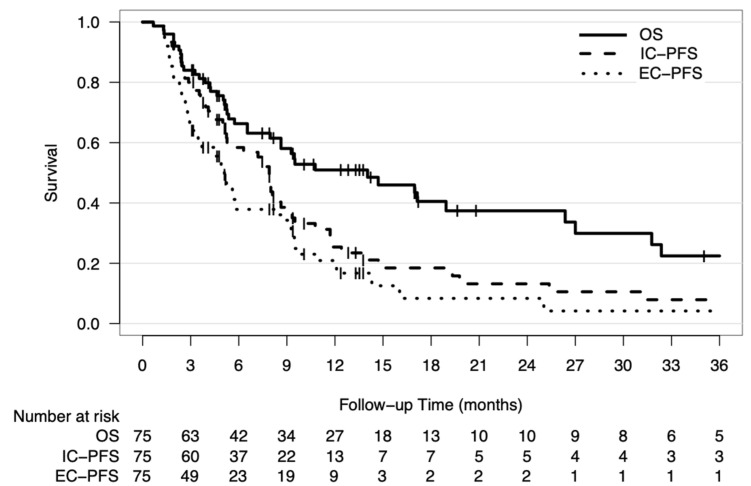
Survival outcomes. Kaplan–Meier curves of overall survival, intracranial progression-free survival (IC-PFS), and extracranial progression-free survival (EC-PFS) for the entire study cohort. Abbreviations: IC-PFS; intracranial progression-free survival, EC-PFS; extracranial progression-free survival.

**Figure 2 cancers-13-00380-f002:**
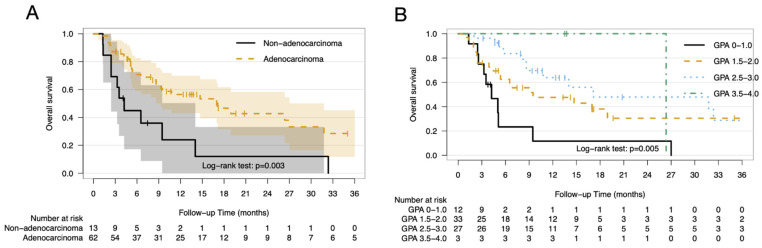
Survival outcomes by histology and GPA. Kaplan–Meier curves of overall survival stratified by histology subtype (**A**) and lung-molecular GPA score (**B**). The shaded area represents the 95% confidence interval. Abbreviations: GPA; graded prognostic assessment.

**Figure 3 cancers-13-00380-f003:**
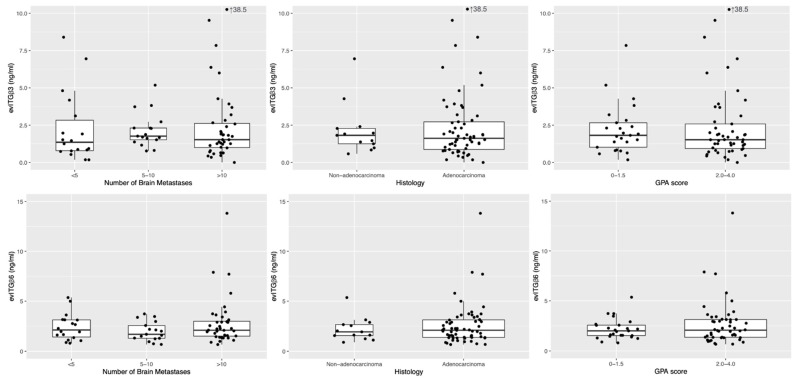
Distribution of extracellular vesicles integrin concentrations. Boxplots and scatterplots showing the distribution of circulating extracellular vesicles integrin (evITG) β3 (upper row) and evITG β6 (lower row) by the number of brain metastases (**left**), histology subtype (**middle**), and GPA score (**right**). Abbreviations: GPA; graded prognostic assessment.

**Figure 4 cancers-13-00380-f004:**
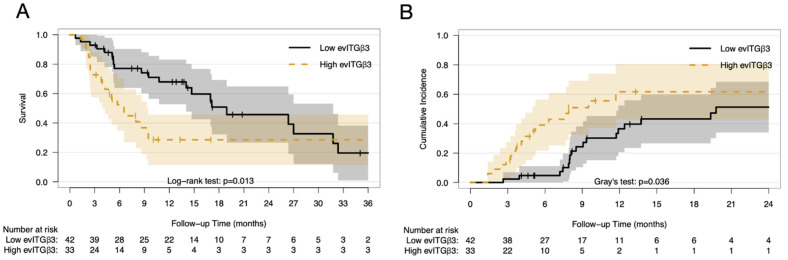
Survival outcomes by circulating extracellular vesicles integrin β3 levels. Overall survival (**A**) and cumulative incidence of intracranial failure (**B**) by dichotomized circulating EV integrin β3 levels. The shaded area represents the 95% confidence interval. Abbreviations: evITG; extracellular vesicles integrin.

**Table 1 cancers-13-00380-t001:** Patient characteristics and clinical outcomes.

Characteristics	Number	%
Number of Patients	75	100%
Median age, years (range)	61 (38–81)	
Gender		
Male	40	53.3%
Female	35	46.7%
Prior SRS	6	8.0%
Number of brain metastases		
<5	18	24.0%
5–10	17	22.7%
>10	40	53.3%
Leptomeningeal seeding	45	60.0%
Histology		
Adenocarcinoma	61	81.3%
Squamous cell carcinoma	4	5.3%
Non-small cell carcinoma, NOS	5	6.7%
Small cell carcinoma	4	5.3%
Adenosquamous	1	1.3%
GPA		
0–1.0	12	16.0%
1.5–2.0	33	44.0%
2.5–3.0	27	36.0%
3.5–4.0	3	4.0%
Median follow-up, months (range)	8.0 (0.7–52.1)	
Death		
Yes	44	58.7%
No	31	41.3%
Intracranial progression after WBRT		
Yes	37	49.3%
No	31	41.3%
Unknown	7	9.3%
Salvage treatment		
Surgery + SRS	1	1.3%
SRS	3	4.0%
SRT	2	2.7%
TKI	2	2.7%
WBRT + Bev	3	4.0%
Extracranial progression		
Yes	45	60.0%
No	23	30.7%
Unknown	7	9.3%

Abbreviations: NOS, not otherwise specified; GPA, graded prognostic assessment; WBRT, whole brain radiotherapy; SRS, stereotactic radiosurgery; SRT, stereotactic radiotherapy; TKI, tyrosine kinase inhibitor; Bev, bevacizumab.

**Table 2 cancers-13-00380-t002:** Multivariate Cox regression analysis of factors associated with overall survival.

Prognostic Factors	HR	95% CI	*p*-Value
Age (per 1-year increase)	1.04	1.00–1.08	0.04 *
Sex (female vs. male)	1.22	0.55–2.72	0.62
Histology (adenocarcinoma vs. non-adenocarcinoma)	0.38	0.18–0.83	0.02 *
GPA score (high vs. low)	0.21	0.09–0.45	<0.001 *
Integrin β3 (per 1-ng/mL increase)	1.25	1.04–1.49	0.02 *
Integrin β6 (per 1-ng/mL increase)	0.90	0.68–1.18	0.44

Abbreviations: HR, hazard ratio; CI, confidence interval; vs., versus; GPA, graded prognostic assessment. Asterisk denotes significance.

**Table 3 cancers-13-00380-t003:** Competing risk regression analyses of factors associated with intracranial and extracranial failures.

Intracranial Failure	sHR	95% CI	*p*-Value
Age (per 1-year increase)	1.021	0.969–1.07	0.44
Sex (female vs. male)	0.817	0.438–1.52	0.52
Histology (adenocarcinoma vs. non-adenocarcinoma)	1.235	0.463–3.30	0.67
GPA score (high vs. low)	2.059	0.823–5.15	0.12
Integrin β3 (per 1-ng/mL increase)	1.216	1.012–1.46	0.037 *
Integrin β6 (per 1-ng/mL increase)	0.868	0.679–1.11	0.26
Intracranial failure (using cut-off points for integrin levels)			
Age (per 1-year increase)	1.033	0.983–1.09	0.2
Sex (female vs. male)	0.833	0.436–1.59	0.58
Histology (adenocarcinoma vs. non-adenocarcinoma)	1.274	0.417–3.89	0.67
GPA score (high vs. low)	1.939	0.758–4.96	0.17
High integrin β3 (>1.818 ng/mL vs. ≤1.818 ng/mL)	2.147	1.106–4.17	0.024 *
High integrin β6 (>1.99 ng/mL vs. ≤1.99 ng/mL)	0.796	0.421–1.51	0.48
Extracranial failure			
Age (per 1-year increase)	1.003	0.968–1.04	0.88
Sex (female vs. male)	0.813	0.437–1.51	0.51
Histology (adenocarcinoma vs. non-adenocarcinoma)	2.013	0.664–6.10	0.22
GPA score (high vs. low)	0.619	0.319–1.20	0.16
Integrin β3 (per 1-ng/mL increase)	1.052	0.830–1.33	0.68
Integrin β6 (per 1-ng/mL increase)	0.974	0.707–1.34	0.87

Abbreviations: sHR, subdistribution hazard ratio; CI, confidence interval; GPA, graded prognostic assessment; vs., versus. Asterisk denotes significance.

## Data Availability

The data presented in this study are available on request from the corresponding author. The data are not publicly available due to privacy or ethical restriction.

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
