# Peer review of "Circulating Exosomal Integrin β3 Is Associated with Intracranial Failure and Survival in Lung Cancer Patients Receiving Cranial Irradiation for Brain Metastases: A Prospective Observational Study"

_cancers, 2021, doi:10.3390/cancers13030380_

Round 1
Reviewer 1 Report
The authors addressed all my concerns appropriately.
Reviewer 2 Report
The authors have answered all the perminent and important questions raised.
This manuscript is a resubmission of an earlier submission. The following is a list of the peer review reports and author responses from that submission.
Round 1
Reviewer 1 Report
This manuscript presents an interesting association of circulating integrin beta-3 with worse prognosis in patients with lung cancer brain metastasis. There are some issues that need to be addressed.
The only major issue is that there is no evidence that the exosomes are actually deriving from cancer cells. A correlation between integrin expression in the primary tumor and in the circulating exosomes, or the correlation of other known tumor markers in the integrin-positive exosome fractions, would increase the confidence in the derivation of the measured integrins from cancer cells.
Minor issues:
Usually, when presenting data from the literature it is appropriate to use present tense as opposed to past tense (lines 66-68).
In the beginning of the Results section, it would be good to mention how many patients were included in the study.
Is there a particular reason why the histology was predominantly adenocarcinoma? This should be commented.
In lines 241-242, the authors state that “circulating exosomal integrin signatures can aid clinical decision-making for patients 241 with BM”. How does the knowledge of increased exosomal integrin translates into a different patient management course? This should be explained better. Do increased exosomal integrins require a more aggressive treatment? This is not clear.
Author Response
Please see the attachment for the detailed point-by-point response.

Reviewer 2 Report
In this manuscript by Guann-Yiing Chen et al. the authors describe that “Circulating Exosomal Integrin β3 is Associated with Intracranial Failure and Survival in Lung Cancer Patients Receiving Cranial Irradiation for Brain Metastases: A Prospective Observational Study”. This is a interesting study with high potential clinical application as they propose the use of a new prognostic biomarker in lung cancer patients with brain metastasis, a group that actually has dismal prognosis. However, there are some critical points that should be addressed.
Major concerns:
- According to the International Society of Extracellular Vesicles (ISEV), which is the largest organization with the best experts in the world on extracellular vesicles and exosomes, they recommended that “unless authors can establish specific markers of subcellular origin that are reliable within their experimental system(s), authors are urged to consider use of operational terms for Extracellular vesicles subtypes” (Théry, C. et al. J. Extracell. Vesicles 7 (2018)). Thus, the authors are encouraged to exchange the term exosomes by extracellular vesicles along the whole manuscript.
- Moreover, according to these rules, for claiming the isolation and analysis of extracellular vesicles the authors need to characterize the nature of their analyzed particles by three methods (Théry, C. et al. J. Extracell. Vesicles 7 (2018). As correctly indicated in the potential limitation number 3, controls for exosome isolation and integrin quantification are needed. First, the authors should characterize their extracellular vesicle samples by electron microscopy, nanoparticle-tracking analysis and western blot for demonstrating size of the vesicles, concentration and presence of extracellular vesicles markers. For the latter, authors need to show western-blot analysis of specific markers such as transmembrane proteins, intracellular proteins associated to membrane proteins, and intracellular proteins non-associated to plasmatic membrane.
- Moreover, controls for integrin quantification are needed. The authors should demonstrate that the obtained integrin β3 levels (concentration per 1ng/ml) derive from their presence on extracellular vesicles and not coming from circulating integrin β3 or from platelet debris that can be co-isolated during extracellular vesicle isolation. One of the examples for this kind of analysis is immunogold electron microscopy.
- The authors need to include a table (could be included as supplementary) with the univariate analysis of overall survival and for competing risk of all clinical variables such as the previous treatment, number of brain metastasis, etc, as some of this variables might be related to the outcome of the patients. Then, they also need to provide the rationale for the inclusion of variables in the multivariate models such as integrin β6 that are not associated with OS and may need to be excluded from this analysis.
Minor concerns:
- The authors need to include the volume of plasma used for their analysis. Moreover, they included the number of patients in the methods, but it would be nice for the reader to easily find this number in the abstract as well as table 1.
- Figure 2: Indicates Overall survival in the figure.
- Line 118-126. Please include this information as a supplementary table.
- Line 134: Sex and gender is not represented in figure 3 so the sentences should be re-written accordingly.
- Line 39 in the abstract. When talking about the potential of integrin β3 as a novel biomarker for BM, specify that it is a prognostic biomarker.
- Six patients had received prior stereotactic radiosurgery to brain metastases, apart from the WBRT, did these patients perform differently or showed different outcome or levels of integrin β3 and 6?
- In table 1: Histology 5 patients are included as Non-small cell carcinoma, indicate the subtype or “other subtypes” to specify that they are different from the rest of NSCLC included in the study.
Author Response

(The authors gave the same response as above.)
